# Finer SHM-Coverage of Inter-Plies and Bondings in Smart Composite by Dual Sinusoidal Placed Distributed Optical Fiber Sensors

**DOI:** 10.3390/s19030742

**Published:** 2019-02-12

**Authors:** Venkadesh Raman, Monssef Drissi-Habti, Preshit Limje, Aghiad Khadour

**Affiliations:** 1PRES LUNAM IFSTTAR CS4 Route de Bouaye, 44344 Bouguenais, France; venkadesh.raman@calcul-meca.fr (V.R.); preshitlimje24@gmail.com (P.L.); 2Components and Systems Department, Université Paris-Est, IFSTTAR, 77420 Champs-sur-Marne, France; aghiad.khadour@ifsttar.fr

**Keywords:** fiber optic sensor, structural health monitoring (SHM), smart composite material, finite element analysis (FEA), strain sensing, numerical modeling

## Abstract

Designing of new generation offshore wind turbine blades is a great challenge as size of blades are getting larger (typically larger than 100 m). Structural Health Monitoring (SHM), which uses embedded Fiber Optics Sensors (FOSs), is incorporated in critical stressed zones such as trailing edges and spar webs. When FOS are embedded within composites, a ‘penny shape’ region of resin concentration is formed around the section of FOS. The size of so-formed defects are depending on diameter of the FOS. Penny shape defects depend of FOS diameter. Consequently, care must be given to embed in composites reliable sensors that are as small as possible. The way of FOS placement within composite plies is the second critical issue. Previous research work done in this field (1) investigated multiple linear FOS and sinusoidal FOS placement, as well. The authors pointed out that better structural coverage of the critical zones needs some new concepts. Therefore, further advancement is proposed in the current article with novel FOS placement (anti-phasic sinusoidal FOS placement), so as to cover more critical area and sense multi-directional strains, when the wind blade is in-use. The efficiency of the new positioning is proven by numerical and experimental study.

## 1. Introduction

Designing a new generation of offshore wind turbine blades is a great challenge as the size of blades is expected to become huge in the near future. As off-shore wind-turbine blades work under harsh environmental conditions, there is a need to increase the efficiency, use safety, strength and reduce the structural weight of offshore wind-turbines [1]. In wind-blades, two different composite structures, named upper composite structure and lower composite structure are manufactured from prepreg materials and later these huge structures are bonded together by an adhesive [2]. As the edges are bonded by adhesives, debonding between the upper and lower composite structures when they are operating is possible. The majority of failures in wind turbine blades occur because of debonding of the adhesives trailing edges [1,2,3]. As trailing edge of the blade is related to serious issues (Figure 1a,b), it is essential to monitor this region with the help of real-time structural health monitoring (SHM) systems that are able to inform about excess deformation and help taking appropriate actions when needed in order to avoid heavy maintenance [3].

SHM technologies, which use embedded Fiber Optics Sensors (FOSs) are incorporated in critical stressed zones [1,4,5,6,7]. FOSs can be embedded within or can be surface-mounted on the composite structure. The embedding of FOSs within the composite structure makes sense as long as the FOSs are small enough to be considered non-intrusive, are naturally noise free and show low electrical risks. Also, they can deliver optimised and precise signals in harsh offshore environments [1,3]. 

In this research work, two important issues are addressed by using numerical simulations and experimental validations. When a FOS is embedded in a composite material it is important to consider that the FOS must be as little intrusive as possible in a way to not influence the mechanical and structural properties of the host material [1,3,6]. The diameter and coating material of the FOS play a vital role when embedding techniques within composite plies are considered. When a FOS is embedded within a composite, a ‘penny shape’ region of resin concentration around the FOS is formed, which diameter depends on the FOS’s size [7,8]. This penny shape region is highly stress concentrated and can lead to delamination at early stages of loading within composites plies. Therefore, attempts have absolutely to reduce as much as possible the sizes of these penny shape regions around the FOS, when embedded in composite plies.

The placement of an FOS within carbon composite plies is the critical issue, which is addressed in this research work. As FOS offers a reliable way to monitor a large structure area of a wind turbine blade, it is therefore essential to optimize the placement of FOS within bonded areas in order to monitor the blades [1]. The previous research work done in this field suggests two FOS alignments (linear and sinusoidal alignment) to monitor the critical zone [1], but the placement for better surface coverage of the critical zone still needs some improvement. Therefore, further advancement is proposed with a novel FOS placement that we call “dual-sinusoidal FOS placement”, so as to cover more critical area and sense multi-parameter strain signals when the blade is in use. Also, when this type of placement is considered, it is important to optimize several parameters in order to avoid early delamination from the host material. The efficiency of the proposed positioning is proven by numerical and experimental studies. 

When the diameter of a FOS increases, more resin concentration is formed around the FOS, which increases the problem of early delamination and can affect the material properties. This intrusive effect due to different diameters of 125 and 250-microns is shown in Figure 2a,b (1). Resin concentration also creates stress concentrated zones in the local area and the study shows that higher stress concentration occurs at the vertices (sharp areas) of the penny shaped region, which is the weakest point where delamination or Mode II crack propagation can easily start [6]. Thus, it is obvious that a smaller diameter, 125 microns, is preferable as the intrusive effect is reduced to a great extent when compared to a core diameter of 250 microns.

## 2. Objectives

Based on previous numerical and experimental results, a few points have been noted and then addressed in this research paper with a novel FOS placement [1]. In the case of single linear alignment of FOS, the FOS is not able to gather the strains associated throughout the cross section surface area of the composite specimen. Thus, a new solution involving multi-linear placement of FOS embedded in composite specimens is proposed (Figure 3) in order to gather the strain signals throughout the cross-section surface area of a composite specimen. Also, when this case is considered it is important to optimize the parameters such as the distance edge-to-FOS and distance FOS-to-FOS (ν1 and ν2) as shown in Figure 3 and Figure 4, in order to avoid weakening the structure. Therefore, three different finite element numerical models are set-up with different parallel spacing between the FOS to analyse the strain coverage and their mechanical behaviour within the host material to overcome the drawback.

To obtain proper strain signals and avoid delamination and cracks within the plies of the composite material specimen, it is important to optimise three important variables (ν1, ν3 and ν4) as indicated in Figure 4.

From the previous study (1) and results with single-sinusoidal placement, it is possible to measure multi-direction strain parameters of the entire surface area of the specimen, but not finely enough. When the impact or compression load is acting in the opposite phase of single sinusoidal alignment, the sensing system is not able to detect the specific loads. To achieve more accurate structural health monitoring, a new dual sinusoidal alignment, embedding two FOS in opposite phase to each other is proposed (Figure 5). A finite element model is set-up where the numerical results are validated by experiments.

## 3. Numerical Modelling

Our study focuses on providing an optimised way of embedding FOS within composite plies for best possible coverage of strain signals for wind-turbine structures and/or similar applications. There are some finite element analyses realized for mechanical and thermal stress studies between FOS and host materials [1,6,7]. Our numerical models are developed using the commercial software ABAQUS. Numerical models include various parts of different materials that are assembled together to compose a final specimen model for finite element simulations. The FOS is designed with the significant coating material properties mentioned in Table 1. Two carbon fiber composites are bonded together with an epoxy matrix, which hosts a linear, multi-linear, sinusoidal, and dual-sinusoidal placed optical fiber. The following table illustrates the material properties used for numerical modelling.

For the dual-sinusoidal model suggested, the FOS is placed length-wise in the specimen, with the sinusoidal peaks as shown in Figure 5. The finite element analysis of this specimen is considered under different mesh conditions. All parts of the specimen are meshed with different global seeds to fulfil the most advanced numerical concentration around the focal point of our analysis. As the specimen model is created with different material properties, one should consider the isotropic and composite material properties during analysis. The first upper and lower layers of the model are composite laminate with a 0° orientation. As for the complexity of the geometry, the model is meshed with structured, free tetrahedral and boundary optimized free tetrahedral meshes. The mesh size is varied for each part of the model. A fine mesh is used around the FOS to avoid calculation errors and distortion.

The numerical model is constructed by using two sinusoidal optical fibers embedded in the composite, where the phases of the two sinusoidal optical fibers are opposite one another. The numerical model comprises various parts, where the first layer is a carbon composite, the second layer is of epoxy where the two optical fibers with coating, cladding and core in a sinusoidal pattern is embedded. The last layer is made of carbon composite which sandwiches the model (Figure 6). 

With the help of this model, the idea was to optimise the three important variables (ν1, ν3 and ν4) and obtain best possible strain signals for coverage of the structure. Material properties assigned are mentioned in Table 1. The tensile load is considered to obtain the strain parameters measured by optical fiber within the model. Load and boundary conditions are therefore similar to the multi-linear FOSs model (1), where uniform pressure is applied on one side of the model and other side is considered fixed (Figure 7). For more details about boundary conditions readers should report to the (1).

## 4. Experimental Preparation

To study the effectiveness of the current proposal of dual FOSs positioning, a experiment was set up for mechanical tests under three-point bending load. A specimen made of glass fiber-reinforced composite material manufactured by pultrusion was considered, with dimensions of 800 × 30 × 20 mm, for the experiments. FOSs of 0.125 mm diameter with 0.08 mm of acrylate coating were then placed on the surface of the glass fiber composite with dual-sinusoidal placement, as discussed. Then bi-component epoxy resin was applied to embed the FOSs. The FOSs are bonded as shown in Figure 8 with equal intervals. The peak is maintained at the middle of the specimen and spread towards the tip of the specimen linearly. For dual-sinusoidal alignment an extended sinusoidal placement of 6-cm period waves was considered because from previous studies, this placement has shown better strain sensing ability when compared to a linear case [1]. To, maintain the optical fibers’ position, they are bonded initially on the working bench, along with the the composite specimens. This helps to avoid any fiber misalignment during bonding. 

Once the bonding is completed at room temperature, all specimens were subjected heating at 40°C, as recommended by the adhesive’s producer. After establishing the bonding of different FOS alignments another composite layer was placed over it and then the mechanical experiments were carried out. Three-point bending tests were carried-out (with a displacement-controlled load) on the middle of the specimen. Also, torsion was applied with a wedge support at the ends of the specimen with a wedge angle of 45° and imposed displacement of 3.5 mm. Strain signals were thereafter monitored continuously with a LUNA OBR-4600 type optical back-scattering reflectometer (LUNA company, Roanoke, VA, USA), when the composite specimen undergoes mechanical loading.

## 5. Results

### 5.1. Numerical Simulation Results 

#### 5.1.1. Results of Numerical Simulation of Multi-Linear Alignment of Optical Fibers Embedded in Composite Material

Numerical models are simulated using static loading conditions by using standard increments, as proposed by the ABAQUS software. For all models, results were mainly monitored in the epoxy region so as to understand the behaviour of FOSs when embedded in composite. The result analysis was carried for three different finite element models, with different spacings between the FOSs when placed linearly. The following are the results of the experiments performed to see the stress variation in the epoxy part when linear FOSs are placed side by side in the composite material. The stress transfer in the S22 direction (lateral direction) has been measured for the epoxy part when FOSs are present between the composite parts. The stress is expressed in MPa units. In Figure 9, the epoxy stress results are displayed when a numerical model having three FOSs in linear placement is considered. The results in the zoomed image show that stress concentration is present in the epoxy part due to the presence of the FOSs and this stress is then further distributed in a lateral direction till the location of the next FOS, and then further carried forward till the edge of the specimen. The stress measured in the red spots around the FOS in the epoxy part is 35.27 MPa, which then gradually decreases to orange and yellow regions with a value between 25.2 MPa to 20.02 MPa, respectively. Similar behaviour was also observed for the other two top and bottom fibers, too.

Similar results were obtained for the model with two FOSs by subtracting the middle FOS from the previous three FOS model. The value obtained in the stress concentration part of the epoxy in the red zone was found to be 35.35 MPa (Figure 10) which is close to the value of the numerical model with three FOS. Also, the regions in between the orange and yellow parts showed values between 25.18 MPa and 20.09 MPa, which is also close to the numerical model with three FOSs. In the third numerical model, similar kind results were obtained even after changing the spacing between two FOSs (Figure 11). The stress concentration noticed near the epoxy part in the red zone was 35.89 MPa, which then gradually decreases to 25.59 MPa to 20.45 MPa in between the orange and yellow zones, but when the nodal values were considered for each of the above three numerical models and plotted for stress values in the lateral direction, a major change was seen in the results.

The graph in Figure 12 indicates the stress concentration calculated by nodal values in the epoxy part for three different numerical models with three FOSs, two FOSs and two FOSs with the change in distance. The plotted nodal values for the different numerical models show that the stress is varying in the epoxy part in a lateral direction due to presence of the FOSs, which can affect the side by side placement of FOSs. 

The green coloured graph shows the stress variation in the numerical model having three fibers. It shows that the stress is maximum (35.27 MPa) at the near FOS in the epoxy and then it decreases with the distance between two adjacent fibers. It is clearly seen that there is a fluctuation of stress values from 22.23 MPa to 20.10 MPa in between the FOSs when they are placed at a distance of 75 mm from each other. Also, the stress value calculated from the edge of specimen to the first FOS, which is at a distance of 75 mm, rises to 19.31 MPa and then settles down around 19 MPa. When the blue graph is considered which corresponds to the numerical model with two FOSs where the distance between the two FOSs is 150 mm, it is seen that maximum stress value around the FOS in the epoxy is 35.35 MPa, but this value decreases to around 18.28 MPa. The graph becomes stable and linear for the stress distribution in the epoxy till the location of the next FOS. As the distance from the edge of third FOS is 75 mm, similar results are obtained when compared to the model with three FOSs. The major change in results is seen for the third model, which is shown by the red graph values, where the distance between two FOS is 100 mm and the distance of the first FOS from the edge is also 100 mm. It is clearly seen that in this case there is no fluctuation in between two FOSs and a stable stress value of 18.18 MPa is obtained. Also, the stress value from the edge of the specimen is lowered to around 18 MPa when compared to other models. These numerical simulations show clearly that the distance between adjacent fibers for placement of multiple optical fibers needs to be optimized in order to avoid high stress concentration that may lead to easy debonding around the FOSs. 

#### 5.1.2. Results of Numerical Simulation of Dual-Sinusoidal Alignment of Optical Fibers Embedded in Composite Material

The following results are for the case of dual sinusoidal FOSs placement. All strain measurements were taken for the optical fiber core and the study was based on boundary conditions similar to the ones which were used for the multi-linear FOSs placement. The result of the finite element model of dual sinusoidal FOSs placement with acrylate coating material is shown in Figure 13. The unit of strain for the following results is mm/mm. Figure 13 explains the overall strain measurements of the fiber optical core of the dual sinusoidal model.

As discussed in previous results of single sinusoidal FOS placement, multi-parameter strain sensing is possible by using a sinusoidal placement, which means that the sinusoidal placement can measure strain signals in the longitudinal as well as the lateral direction, so an attempt was made to cover the maximum surface area of the specimen to measure multi-parameter strains. From the results it can be seen that the dual-sinusoidal model can sense strain signals and can cover the maximum surface area for fine SHM (placement in dual-sinusoidal mode, in phase opposition, provides coverage complementary to that of a single sinusoidal fiber) and almost the entire surface of the structure can be monitored. The red zone on the peak of both optical fibers in the figure above marks the maximum strain value of 0.03920 which gradually decreases to 0.00381 in the blue zone of the optical fiber. A variation is observed at the end sections of the model, which is due to the applied load and boundary conditions.

The graph of Figure 14 illustrates the nodal strain value variation calculated within the core of the optical fiber. When the above two graphs are compared it can be seen that both optical fibers are able to provide strain values respective to their coordinates, as presented in Figure 6 and Figure 7. Strain values located to the right of the graph belong to sinusoidal optical core 1 and strain values located at the left side indicate the strain value for the sinusoidal optical core 2. It can be seen that strain measurement is possible with the help of dual-sinusoidal FOSs placement by covering the maximum surface area when it is subject to mechanical loading. The slight change at the ends of the graph is due to the applied boundary conditions. To verify this section, experimentation was carried-out and results are mentioned in the following experimental results section.

### 5.2. Experimentation Results

It is important to remember that the experiments conducted are not meant to be validations of the numerical models. Through experimentation, it is simply a question of showing that the hypotheses of the models are coherent and lead to realistic results. Distributed FOS placements are studied to see the effectiveness of sensor positioning for their sensitivity, the elastic range of measurements and multi-parameter measurements (strain at bending and torsion). It is assumed that linear alignments of sensors are able to observe lateral stresses, but not sensitive to longitudinal and shear loads, from previous numerical simulations [1]. Therefore, sinusoidal positioning is considered as a better solution for various directions of strain sensing and also for large surface area coverage using a single FOS. For better coverage of the specimen, a dual-sinusoidal alignment is considered. The graph (Figure 15) shows bending strain measurements carried out with a dual-sinusoidal FOSs alignment on the glass fiber composite specimen. The value measured by both FOSs during bending tests is observed. FOSs are in opposite phase to each other, but it is seen that both fibers are able to measure strain signals. It has been observed that in the strain measurements under bending (when comparing one fiber to the other), the strain value calculated is around 1200 µm/m for both fibers [1]. This clearly explains that dual-sinusoidal placement can be used efficiently for strain measurements. The technique suggested can cover the maximum surface area of the specimen, too. Although this result seems promising, one should absolutely keep in mind that additional experiments have to be conducted under various loading conditions and modes to exhaustively convince users. 

## 6. Discussion

The comparison between multi-linear and dual-sinusoidal FOSs placement is made with the help of numerical simulation and experiments. This study focuses on the strain sensing abilities of different fiber placements when embedded in composite materials. In our study carbon fiber-reinforced composite material is considered for the numerical study and glass fiber-reinforced composite for the experimental method. The change in composite materials affects the variation in strain value in the numerical and experimental methods, but we are interested in strain components not the strain values during the comparison. The strain values are compared while the same materials are in use (comparison within the numerical model and comparison within experiments). 

Based on the above results, it has been observed that a better possible coverage of surface is obtained by dual-sinusoidal FOSs alignment, as it can measure the multi-direction strain signals then compared to multi-linear FOSs alignment, as this alignment is able to sense strain signals only in lateral direction. But, when the multi-linear FOSs placement is used, it possible to find the safe distance and spacing between two parallel optical fibers, which are defined by two variables ν1 and ν2. Thus, one can write the equation for the multi-linear FOSs alignment for rectangular structure, which is given by:(1)$$b=2*v_{1}+\left(n-1\right)*v_{2}$$
where *ν*_1_ is the edge gap, *ν*_2_ is the distance between two optical fibers and *b* is the breadth of the rectangular structure. From Figure 15, the safe variables are obtained as *ν*_1_ = 100 mm and *ν*_2_ = 100 mm for our reduced model (where *n* represents the number of fibers). As from the multi-linear FOSs, it is possible to define the edge gap variable (*ν*_1_) which can be used to define the equation for sinusoidal placement as from the previous single sinusoidal FOSs placement. The other parameters ν_3_ and *ν*_4_ are defined by experimental results. The following equations can be defined for the sinusoidal placement of optical fibers in rectangular structure:(2)$$b=2*v_{1}+2*v_{3}$$
(3)$$l=n*v_{4}$$
where *ν*_1_ is the edge gap, *ν*_4_ is the pitch of sinusoidal curve and *ν*_3_ is the peak of the curve. Also, *b* is the breadth and *l* is the length of the rectangular structure. The defined values for these variable are found to be *ν*_1_ = 100 mm, *ν*_3_ = 30 mm and *ν*_4_ = 30 mm. Based on the calculation above, it is therefore worthwhile to keep in mind that with the optimization (through the equations found) of the variables *ν*_1_, *ν*_2_, *ν*_3_ and *ν*_4_, one can design placement of dual-sinusoidal sensors in a way to avoid any excessive stress-concentration, while conserving full coverage of the surface that researchers and engineers want to monitor. Of course, the placement geometry that was suggested does not completely solve the problem. One of the main drawbacks to it is the local over-strain that can arise from the intersection between the two sinusoids and we are currently thinking about this point.

As a summary of all the numerical models and validation of this work, the most important point to remember is the possibility of using distributed fiber optic sensors in all structures made of composite materials belonging to various industries, whatever their shape, flat or curved. These sensors give the full measure of their physical means without becoming “defects” for the structures, their sizes should be as small as possible, their distance from the edges should be carefully selected in order to avoid too much of a delamination tendency (the parameters *ν* and associated equations of this article can be a source of inspiration for that) and finally, the placement of the sensors must be the object of a reflection and a particular design so that the fineness of the measurements by sensors are the most precise possible, which will allow an optimal coverage of the composite structure considered.

## 7. Conclusions

Structural health monitoring by FOSs that are embedded within a structure is a reliable way for fine SHM. It is worthwhile to note that appropriate placements of FOSs are important in order to sense multi-direction strain parameters and to allow the best surface coverage of the structure. Two FOS placements were considered and analysed to obtain the best surface coverage of the structure and sense strain signals. For multi-linear FOSs alignment, it has been seen that linear placement needs many optical fibers placed parallel to each other, so that it can cover the maximum surface of the specimen. Also, multi-linear placement is not efficient as it can only sense the strain signals in a lateral direction, and though the approximate safe distance for the parallel optical fibers placements is identified, it is not advisable.

On the other hand, the dual sinusoidal FOSs placement is more efficient as it can provide strain signals in the longitudinal as well as lateral directions and thus help measure the bending and torsional loads at the same time. Also, dual-sinusoidal FOSs placement provides the maximum surface coverage of the composite specimen, which is more efficient than multi-linear FOSs placement. Thus, this research work suggests the use of dual-sinusoidal FOSs placement, as it can measure multi-direction strains. The signals also reflect the shape of the sensor; therefore, the study can be helpful for multi-parameter sensing purposes.

Future work will consider the dual-sinusoidal FOSs placement for numerical simulation purposes. It has been assumed that overlapping of two optical fibers is not happening, but in reality, when two optical fibers are placed in a sinusoidal alignment in opposite phase to each other overlapping occurs, which can result in stress concentration, so this aspect must be considered for further work in this research area.

## Figures and Tables

**Figure 1 sensors-19-00742-f001:**
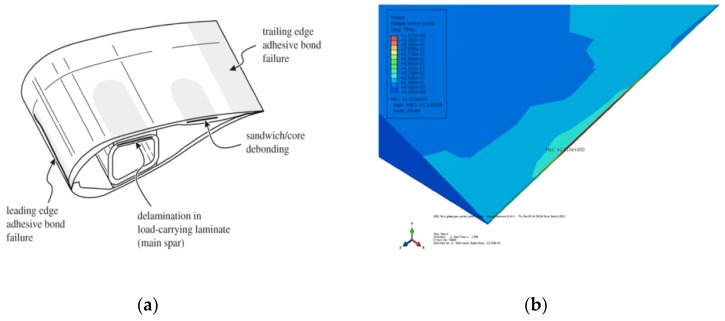
Wind turbine blades - (**a**) Adhesive bonding zones in blade - Critical zones of failure in blades, (**b**) Debonding zone observed at the trailing edge [1].

**Figure 2 sensors-19-00742-f002:**
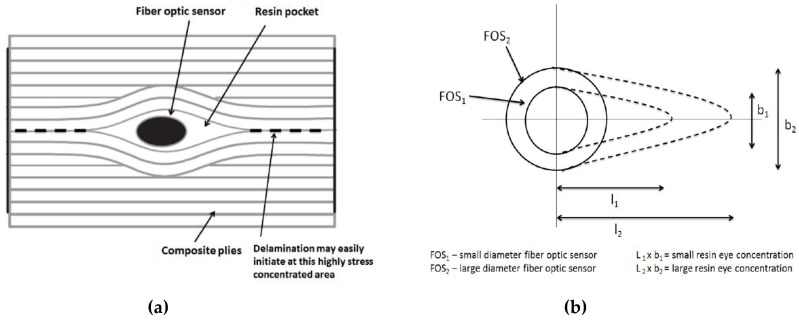
Different resin eye concentrations depending on the diameter and their effect—(**a**) Resin eye concentration around FOS position in composite, (**b**) Resin eye concentration around various diameter FOSs.

**Figure 3 sensors-19-00742-f003:**
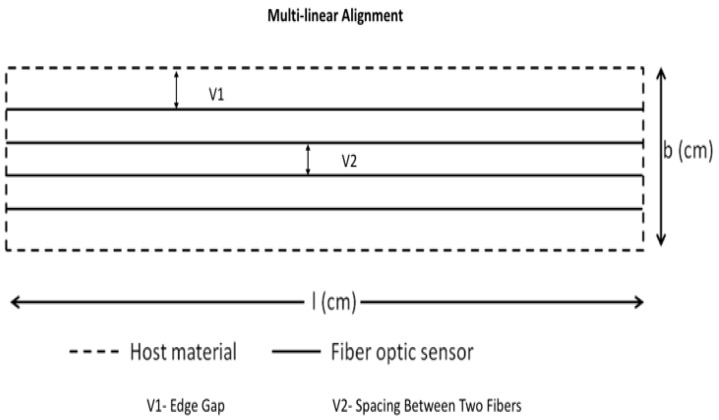
Multiple FOSs embedded in linear alignment showing variables to optimise.

**Figure 4 sensors-19-00742-f004:**
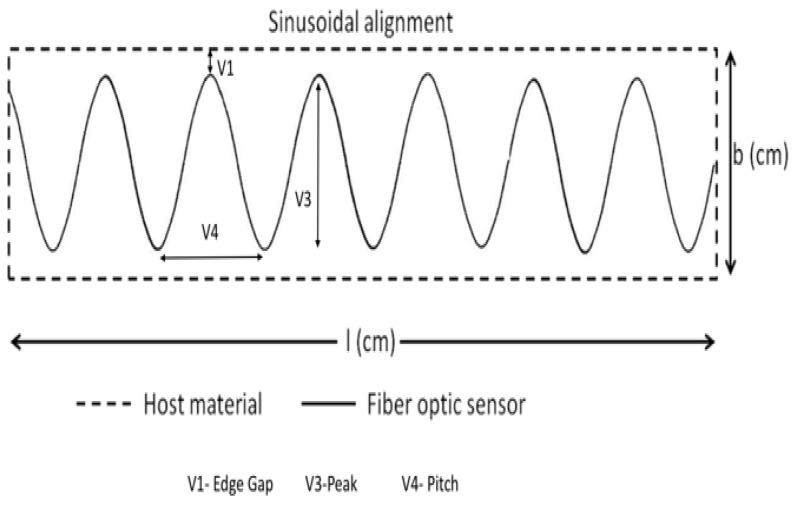
Sinusoidal FOS Alignment Showing 3 variables to optimise.

**Figure 5 sensors-19-00742-f005:**
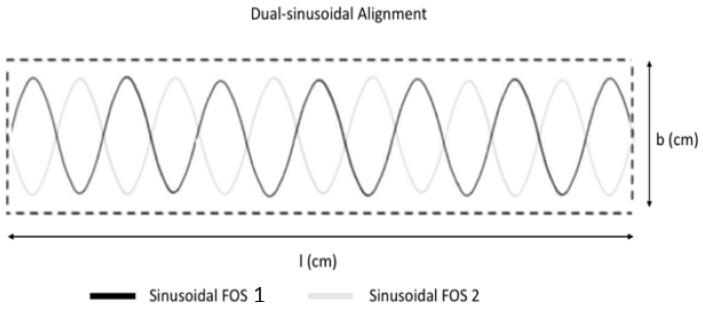
Dual-Sinusoidal FOS Alignment.

**Figure 6 sensors-19-00742-f006:**
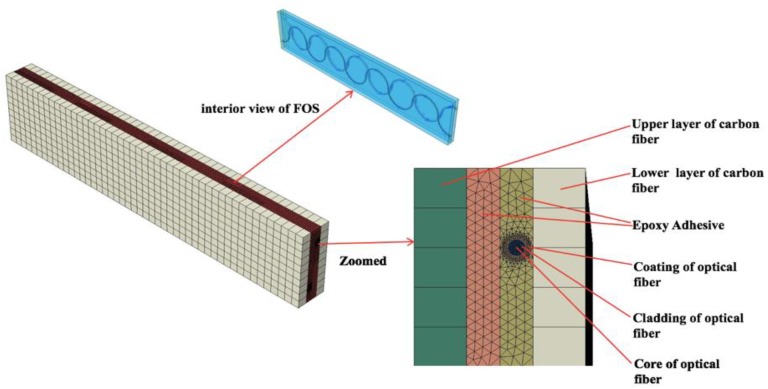
3D and zoomed 2D view of model having dual-sinusoidal optical fiber alignment.

**Figure 7 sensors-19-00742-f007:**
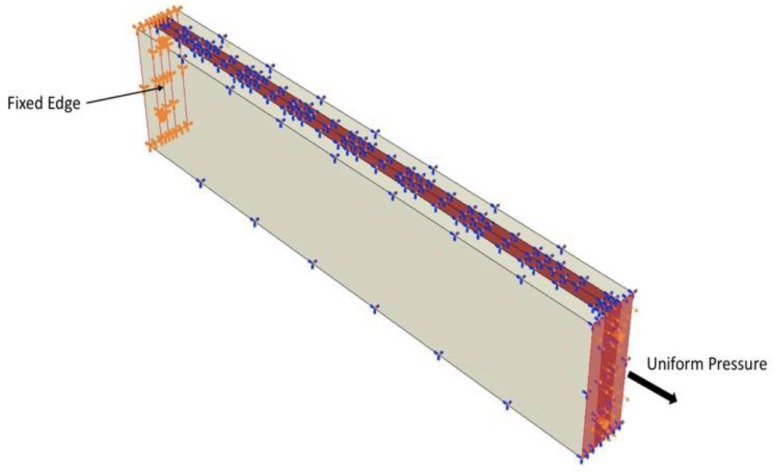
Loads and boundary conditions properties used for model having dual-sinusoidal optical fiber placement.

**Figure 8 sensors-19-00742-f008:**
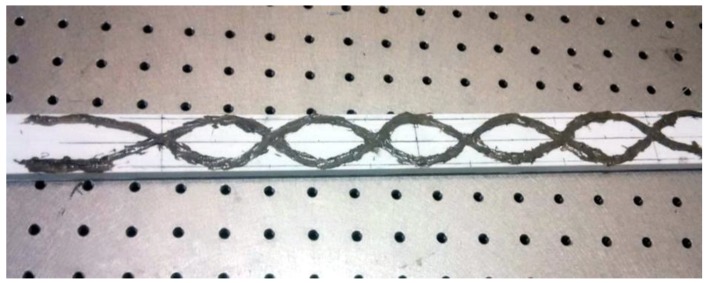
Dual-sinusoidal FOSs’ placement embedded on a glass composite specimen.

**Figure 9 sensors-19-00742-f009:**
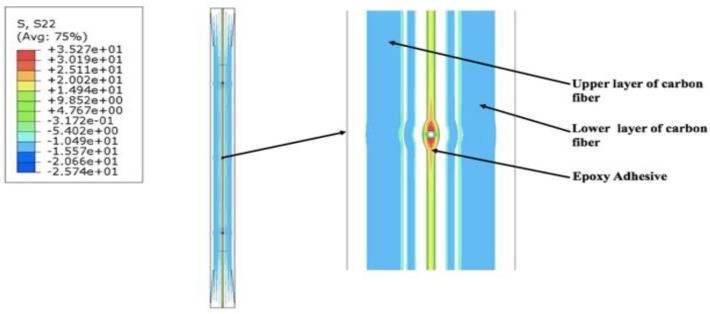
Stress distribution in the numerical model simulation having three FOSs in a linear placement.

**Figure 10 sensors-19-00742-f010:**
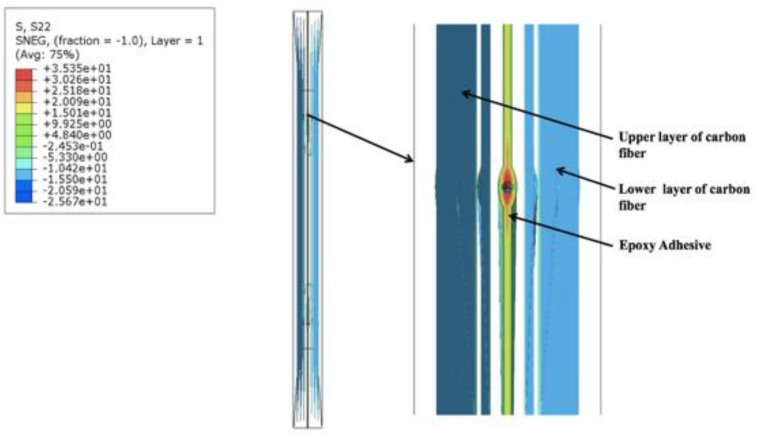
Stress distribution in the numerical model simulation having two FOSs in a linear placement.

**Figure 11 sensors-19-00742-f011:**
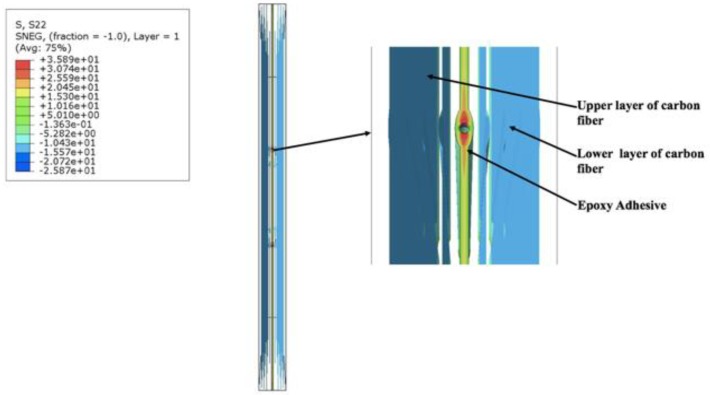
Stress distribution in the numerical model simulation with two FOSs in a linear placement with change in spacing.

**Figure 12 sensors-19-00742-f012:**
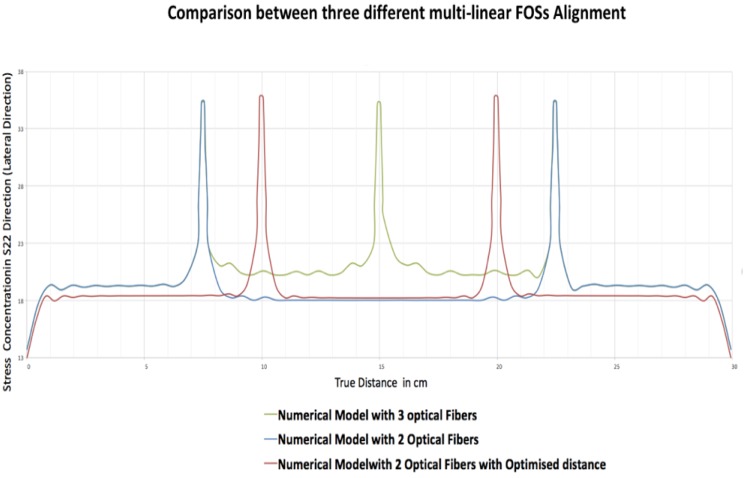
Comparison between three different multi-linear FOSs placements.

**Figure 13 sensors-19-00742-f013:**
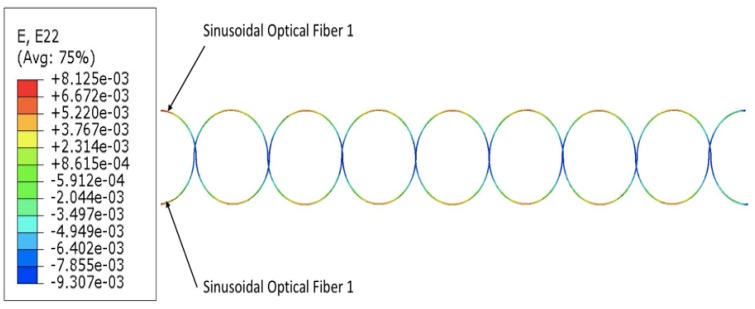
Strain measurements showing dual-sinusoidal FOSs placement (placement in dual-sinusoidal mode, in phase opposition, provides coverage complementary to that of a single sinusoidal fiber).

**Figure 14 sensors-19-00742-f014:**
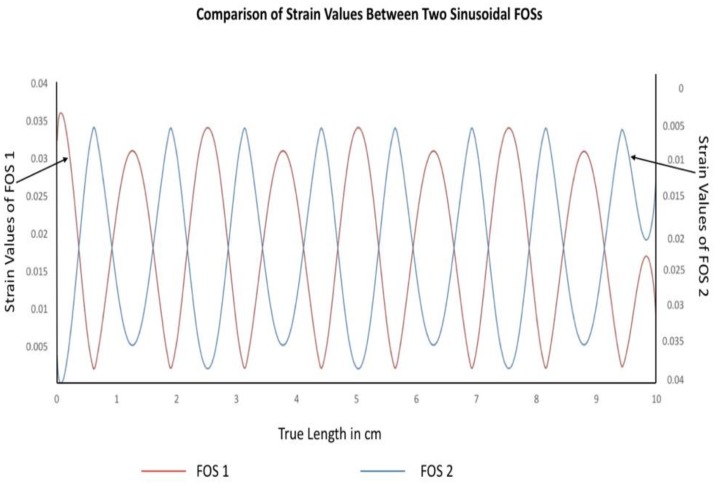
Comparison between strain parameter sensed with dual sinusoidal alignment.

**Figure 15 sensors-19-00742-f015:**
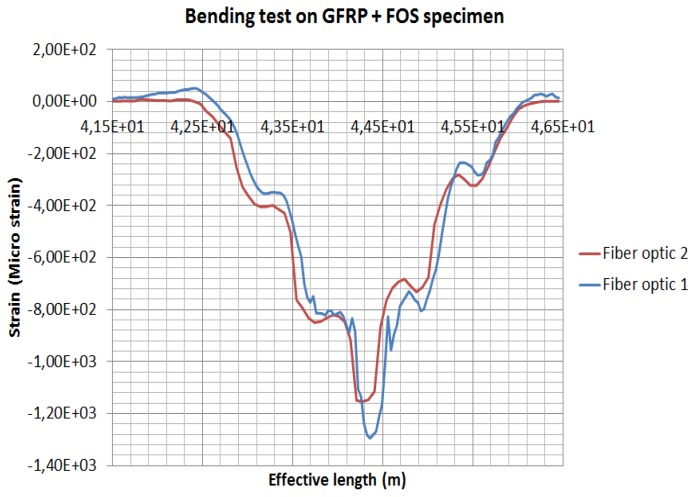
Bending strain measurement carried out with dual-sinusoidal FOSs alignment on glass-fiber composite specimen.

**Table 1 sensors-19-00742-t001:** Mechanical properties of the materials used [1].

Materials	Density (kg/m^3^)	Modulus (MPa)	Poisson’s Ratio
Carbon composite (CFRP)	1950	*E*_1_ = 103,000, *E*_2_ = 10,400, *G*_12_ = 54,000	ν_12_ = 0.3, ν_21_ = 0.03
Epoxy	1250	3500	0.3
Acrylate	950	2700	0.35
Polyimide	1100	3000	0.42
Silica glass	2400	72,000	0.17

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
