# Peer review of "Finer SHM-Coverage of Inter-Plies and Bondings in Smart Composite by Dual Sinusoidal Placed Distributed Optical Fiber Sensors"

_sensors, 2019, doi:10.3390/s19030742_

Round 1
Reviewer 1 Report
This paper reports a dual sinusoidal fiber optics sensors (FOSs) alignment which is more efficient for multi-direction strain sensing. The presented technique can realize simultaneous measurement of strain signals in longitudinal and lateral directions. What’s more, the dual-sinusoidal FOSs placement provides with the maximum surface coverage of the composite specimen. The presented work is technically very interesting and seems appealing for sensing and characterization applications. The paper however is still not ready in its current form. Revisions are needed in order to match the standers of the journal. I personally recommend the authors to go through writing style issues and the grammar problems. Please find below some comments and corrections needed to be addressed by the authors.
The title should be more brief and concise.
The content of this paper is extremely long and too detailed for a research paper, which covers an entire 20 pages. For instance, the introduction almost takes up 5 pages and excessive figures from the previous works are cited with unclear presentations. In my opinion, the writing style is more suited to a doctoral dissertation. Furthermore, it is not beneficial for the readers to grasp the core and essence of the work. Therefore, I strongly recommend the authors to compress greatly the length of the text. Maybe it is better to be less than 10 pages include the expansion of experiment and analysis of the experimental results.
The choice of boundary conditions is very important for simulation. In order to make readers easier to understand the work, it is better to describe the boundary conditions in a much more detailed way.
The dual-sinusoidal FOSs placement can realize multi-parameter strain sensing, I am wondering whether it affects the strain magnitude.
There are lack of the detail of experiment and the comprehensive description of the fig. 6.8. The experimental data current are not enough to support the numerical model.
The statement “multi-parameter strain sensing” in abstract and under Fig. 6.6, as I think, is not standard and strict. Maybe “multi-directional strain sensing” or “multi-point strain sensing” is more suitable.
Most of the figures in this paper would be more easily observed if the font size on the axis is increased and the lines in the figures appear in bold.
The author should check carefully the grammar and text errors in this paper. For example, N/mm2 is not right. In Page 13, the sentence “which has change is spacing between FOSs when placed linear” is also not rightly expressed in the paper.
Author Response
Dear Reviewer
All your comments have been taken into account. Please find the detailed response in the attached file.
Thank you very much

Reviewer 2 Report
- The descriptions in the figures are illegible,
- Descriptions in the figures have a different style
- descriptions have different type of fonts
Author Response
Dear Reviewer,
All your comments have been taken into account.
Best regards
Reviewer 3 Report
This manuscript wants to introduce a new alignment of distributed fiber sensor on the turbine blades by dual sinusoidal shape. The topic is of interest, however, this manuscript is poorly written and hard to follow. It is far away from a journal quality paper. It is formatted more like a report instead of a journal paper. Detail comments for consideration are as below:
1) Introduction repeated a lot of abstract contents and has no new information compare to abstract. Literature review is not sufficient for background.
2) Literature review include so much details from a published paper from authors' research group which were already published such as figures. The repeated information should be summarized and included in introduction instead of repeating a published paper.
3) It claimed in the objective, it is a novel alignment, though it is a new alignment, how novel it is which is doubtful. There is a section 3.1 in the objective, but no section 3.2 at all, why that section is needed? Also, it is presented there are two alignments, multiple linear and dual sinusoidal for investigation, however, in objective, it repeatedly mention single sinusoidal alignment.
4) It modeled three alignments, single linear, multiple linear and dual sinusoidal, however, in experiment, only dual sinusoidal alignment was tested. Also, it is unclear why a single linear is needed to be modeled.
5) There is no comparison in between simulation and experiments.
6) The dual sinusoidal is aligned in perfect circle in simulation but in experiments, it is not perfect circle shape.
7) The results analyzed was very limited. It stated that the dual sinusoidal can detect strains, for sure, it can detect strains. However, how to use these strains and for what purpose, there is no descriptions on any.
8) There is limited information which is useful based on the manuscript, most of the contents are not related.
Author Response
Dear Reviewer
All your comments have been considered. Please find the detailed response in the attached file.
Thank you very much
Best regards

Round 2
Reviewer 1 Report
This paper reports dual sinusoidal placed distributed optical fiber sensors for structural health monitoring in smart composite. Through simulation analysis and experimental study, the results reveal that the dual sinusoidal arrangement can cover more critical area and sense multi-directional strains signals, compared to multi-linear alignment. The experiments and results are well arranged and described. I think that this paper can be considered for publication in Sensors after some necessary revisions as following:
1. The abstract is tortuous rather than refined. The significance of research is described a lot, but the research contents and results are not concrete.
2. In my opinion, Figure 1, Figure 2 and Figure 3 could be deleted since the introduction of a paper usually does not contain images.
3. In introduction, the first passage is lengthy without getting into the point. Additionally, it seems as if there is no logic relation between the second last passage and the last passage in introduction.
4. The fonts in most of figures in the manuscript are too small to be observed clearly. For example, Figure 4 and Figure 6. In addition, Figure 6.7 should be changed into Figure 6.5. Figure 6.8 should be replaced by Figure 6.4. Figure 6.4 cannot be found in the paper.
5. There are still numerous grammar and text errors in the revised manuscript. The authors should check scrupulously and modify them. To provide some examples,
D In abstract, the phrase "Designing of new generation offshore wind turbine blades "might be replaced with "Designing a new generation of offshore wind turbine blades", and the word "dictates" in "which dictates the parameter strength-to-weight ratio to be taken care off " seems improper. The sentence "The size of so-formed defects are depending on diameter of the FOS. Penny shape defects depend of FOS diameter" is repetitive and has syntax errors.
D Line 64, page 2, the expression of sentence "The embedment of FOS within the composite structure makes sense as long as FOS are small enough to be considered not intrusive......" should be rewritten.
D Line 83 or in other locations, the word "multi-parameter strain sensing" might be better if it is replaced with "multi-point/ multidirectional strain sensing".
D In Section 2, line 1-2, the sentence "Based on previous numerical and experimental results, few points to improvement are noted and then addressed in this research paper with novel FOSs placement [1]. " is puzzling. Please check it.
D In line 1-2, Section 2, Fig 2.6 is inexistent.
D Page 5, Passage 1, line 2, Table 3.5 cannot be found.In Section 6.12, passage 2, line 8, "due" should be replaced by "due to"
Author Response
Dear Reviewer,
Thank you very much for the valuable comments. My answers are included in your text, in green.
Best regards
1. The abstract is tortuous rather than refined. The significance of research is described a lot, but the research contents and results are not concrete.
The abstract was shortened as requested.
2. In my opinion, Figure 1, Figure 2 and Figure 3 could be deleted since the introduction of a paper usually does not contain images.
The figures mentioned are important for the rest of the text. They are expected to bring more explanations that will enable readers to understand the results in easier way ...
3. In introduction, the first passage is lengthy without getting into the point. Additionally, it seems as if there is no logic relation between the second last passage and the last passage in introduction.
The introduction has been shortened ...
4. The fonts in most of figures in the manuscript are too small to be observed clearly. For example, Figure 4 and Figure 6. In addition, Figure 6.7 should be changed into Figure 6.5. Figure 6.8 should be replaced by Figure 6.4. Figure 6.4 cannot be found in the paper.
The figures have been magnified
5. There are still numerous grammar and text errors in the revised manuscript. The authors should check scrupulously and modify them. To provide some examples,
D In abstract, the phrase "Designing of new generation offshore wind turbine blades "might be replaced with "Designing a new generation of offshore wind turbine blades", and the word "dictates" in "which dictates the parameter strength-to-weight ratio to be taken care off " seems improper. The sentence "The size of so-formed defects are depending on diameter of the FOS. Penny shape defects depend of FOS diameter" is repetitive and has syntax errors.
The changes have been done
D Line 64, page 2, the expression of sentence "The embedment of FOS within the composite structure makes sense as long as FOS are small enough to be considered not intrusive......" should be rewritten.
Changes have been done
D Line 83 or in other locations, the word "multi-parameter strain sensing" might be better if it is replaced with "multi-point/ multidirectional strain sensing".
The wording "multi-parameter strain sensing" has been used in our previous paper and shopuld be kept in the current to show readers that current work is a continuation ...
D In Section 2, line 1-2, the sentence "Based on previous numerical and experimental results, few points to improvement are noted and then addressed in this research paper with novel FOSs placement [1]. " is puzzling. Please check it.
Changes have been done
D In line 1-2, Section 2, Fig 2.6 is inexistent.
Changes have been done
D Page 5, Passage 1, line 2, Table 3.5 cannot be found.In Section 6.12, passage 2, line 8, "due" should be replaced by "due to"
Changers have been done
Reviewer 3 Report
Comments 5 (experiment and simulation should be compared), 6 (simulation should be modified to be same as experiments), 7 (more analysis is needed to compare why dual alignment is needed), and 8 (contributions need to be clarified through the comparison between different alignments to see why dual alignment is needed) were not addressed in the revised manuscript.
Author Response
Dear Reviewer,
Thank you very much for the very valuable comments. My answers are hereafter.
Best regards
Comments 5 (experiment and simulation should be compared), 6 (simulation
should be modified to be same as experiments), 7 (more analysis is
needed to compare why dual alignment is needed), and 8 (contributions
need to be clarified through the comparison between different alignments
to see why dual alignment is needed) were not addressed in the revised
manuscript.
It is important to remember that conducted experiments are not meant to be validations of numerical models. Through experimentation, it is simply a question of showing that the hypotheses of the models are coherent and lead to realistic results.
Linear placement is not technologically beneficial ... Engineers and researchers need tensile, compressive and torsional strains to be recorded. Linear placement is bringing tensile strains only.
Placement in dual-sinusoidal mode, in phase opposition, provides surface coverage complementary to that of a single sinusoidal fiber
Round 3
Reviewer 3 Report
The paper is revised according to my previous comments.